# YKL-40 and the Cellular Metabolic Profile in Parkinson's Disease

**Maria Gevezova [1,2,†], Maria Kazakova [1,2,*,†], Anastasia Trenova [3,4] and Victoria Sarafian [1,2]**

[1] Department of Medical Biology, Medical University, 4000 Plovdiv, Bulgaria; mariya.gevezova@mu-plovdiv.bg (M.G.); victoria.sarafian@mu-plovdiv.bg (V.S.)
[2] Research Institute at MU-Plovdiv, 4000 Plovdiv, Bulgaria
[3] Department of Neurology, Medical University, 4000 Plovdiv, Bulgaria; anastasiya.trenova@mu-plovdiv.bg
[4] University Hospital "Kaspela", 4000 Plovdiv, Bulgaria
[*] Correspondence: mariya.kazakova@mu-plovdiv.bg; Tel.: +897-571-982
[†] These authors contributed equally to this work.

**Abstract:** Parkinson's disease (PD) is the second most common neurodegenerative disease worldwide. A growing body of evidence suggests that mitochondrial dysfunction and inflammation play a crucial role as a pathogenetic mechanism in PD. The glycoprotein YKL-40 (CHI3L1) is a potential biomarker involved in inflammation and tumor processes. The aim of the present study was to investigate the metabolic profile of PBMCs from PD patients and to search for a possible relationship between cellular bioenergetics and YKL-40. The study included 18 naïve PD patients and an age-matched control group (HC, n = 7). Patients were diagnosed according to the MDS-PD, the UPDRS, and the Hoen–Yahr scales. Mitochondrial activity was measured by a metabolic analyzer on isolated PBMCs from PD patients. Gene (qPCR) and protein (ELISA) expression levels of YKL40 were investigated. New data are reported revealing changes in the mitochondrial activity and YKL-40 levels in PD patients. Bioenergetic parameters showed increased respiratory reserve capacity in PD compared to HC. The protein levels of YKL-40 were threefold higher in PD. We found a correlation between the YKL-40 protein levels and basal respiration and between YKL-40 and ATP production. These observations suggest an interplay between YKL-40 and mitochondrial function in PD. We assume that the YKL-40 gene and protein levels in combination with changes in mitochondrial function might serve as an additional tool to monitor the clinical course of PD.

**Keywords:** Parkinson's disease; YKL-40; mitochondrial dysfunction

## 1. Introduction

Parkinson's disease (PD) is defined as a complex neurodegenerative disorder characterized by dopaminergic neuronal cell death in the substantia nigra [1]. It is the second most common neuropathological condition after Alzheimer's disease, associated with significant disability and poor quality of life [2]. Many factors such as aging, smoking, and alcohol consumption are thought to be related to the initiation and progression of PD [3,4].

It has been suggested that the effect of the aging process on the pathogenesis of PD is partially mediated by mitochondrial dysfunction [5]. With aging, mitochondria become sensitive to environmental factors, thus increasing the possibility of mitochondrial dysfunction. A variety of mtDNA mutations lead to a reduction in ATP production, augment free radicals and oxidative stress, and cause mitochondrial dysfunction [6]. As mitochondrial respiration is linked to ATP production, neurons and differentiated neuronal cells are dependent on oxidative phosphorylation and display a considerable mitochondrial respiratory capacity to meet the requirements of increased functional demands and stress [7]. Strong evidence for altered mitochondrial metabolism in human neurodegenerative diseases, especially in PD patients, is still missing.

The implementation of reliable secreted biomarkers associated with motor or cognitive PD subtypes could allow accurate prediction of clinical outcomes. Recently, the research on novel biomarkers in PD is expanding, and data about prognostic cerebrospinal fluid (CSF) and blood biomarkers have been reported. They could help clinicians reliably predict patients' survival and determine the best choice of therapy. Furthermore, the potential advantage of biomarkers as drug targets in clinical trials has also been discussed.

Chitinase-3-like protein 1 (CHI3L1/YKL-40) or human cartilage glycoprotein-39 is a 40-kDa glycoprotein, a member of the mammalian chitinase-like protein family. It is considered as a marker for macrophage and microglial differentiation and activation [8]. It is an extracellular matrix glycoprotein belonging to glycoside hydrolase family 18 [9]. Because of the amino acid substitutions in the active site of chitinases, YKL-40 has no catalytic activity against chitinase substrates [10]. The protein is normally expressed by multiple cell types such as macrophages, chondrocytes, and vascular smooth muscle cells. However, increased YKL-40 levels have been detected in several solid tumors and chronic inflammatory conditions [9,11,12].

Elevated CSF YKL-40 levels were reported in different infectious and noninfectious diseases of the CNS [13]. Previous studies found that the YKL-40 protein may have a potential role as a promising biomarker reflecting the severity of inflammation in PD [14]. There are no data available on YKL-40 levels and mitochondrial metabolism in PD patients.

Due to the complex nature of this disease, biomarkers with diagnostic and prognostic value are needed to reliably identify the early stages of PD and to stratify patients into clinical subtypes, to allow individualization and the more effective application of therapy. In this context, the present study aimed to search for a possible relationship between cellular bioenergetics and YKL-40 gene and protein levels.

## 2. Results

### 2.1. Patients

Eighteen, newly diagnosed according to MDS-PD criteria, treatment naïve patients (nine women, nine men), aged between 42 and 79 years (mean age $65.5 \pm 11.87$) were included in this study. The duration between the starting point of the symptoms, reported by the patient, and the time point of diagnosis varied between 3 months and 15 months (mean $9 \pm 6.43$ months). The UPDRS score was between 20 and 48 (mean $33.39 \pm 8.261$). According to the Hoen–Yahr scale, the degree of disability in the majority of the patients (15/18, 83.33%) was 1.5 (unilateral signs and involvement of the axial muscles), and 3 out of 18 (16.67%) had 1.0 (unilateral Parkinsonian signs only). The results from both scales indicated that all patients were in the early stage of the disease.

### 2.2. Mitochondrial Activity

After measuring the mitochondrial function in PD patients and healthy controls (HC), we obtained data on five metabolic parameters that best described different factors influencing the ability of PBMCs to meet cell energy needs.

Our results showed an increased spare respiratory capacity in PD patients compared to HC. This difference was statistically significant ($p = 0.038$) and depicted the characteristic respiration pattern of PD PBMC (Figure 1). The studied indicator in patients averaged $243\% \pm 1.15$, while for HC, it was $168\% \pm 0.4$. However, the rest of the bioenergetic data showed a tendency for decreased values in PD. The difference in ATP-coupled respiration ($p = 0.194$), coupling efficiency ($p = 0.161$), maximal respiration ($p = 0.680$), and proton leak ($p = 0.445$), compared to HC, was not statistically significant. In addition, both PD PBMCs and HC PBMCs had a similar mean basal respiration without any statistical significance ($p = 0.794$), indicating the homogenous onset and the same conditions in studying the target groups. The results are shown in Figure 1.

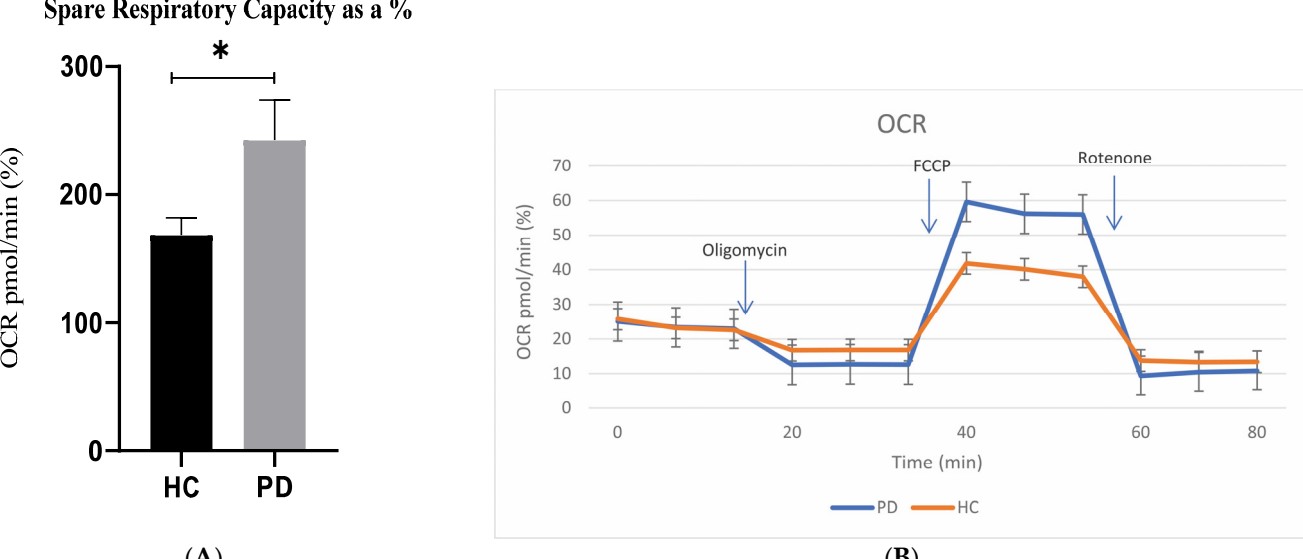

(A)                                                                                          (B)

**Figure 1.** PBMC bioenergetic data in real time (Seahorse Xp analyzer). (**A**) The graph shows the differences between Parkinson's disease (PD) and healthy controls (HC). Values are expressed as means ± S.E.M; * = $p < 0.05$. (**B**) Profiles of Mito Stress Test data for OCR with arrows indicating injections into the media of specific inhibitors: oligomycin, carbonyl cyanite-4 (trifluoromethoxy) phenylhydrazone (FCCP), and Rotenone.

### 2.3. Gene and Protein Expression of YKL-40

The expression levels of YKL-40 showed only a trend for decreased mRNA levels in patients compared to the controls without any statistical significance ($p = 0.651$) in contrast to the protein levels ($p = 0.0001$) (Figure 2). The mean YKL-40 mRNA values in PD were 1.125 ± 0.6, while in HC they were 1.317 ± 0.4. Interestingly, YKL-40 plasma levels in patients with PD were three times higher than those of the HC. The reported mean values for protein levels in PD were 164.70 ng/mL ± 79.7, compared to the control group—46.06 ng/mL ± 19.1.

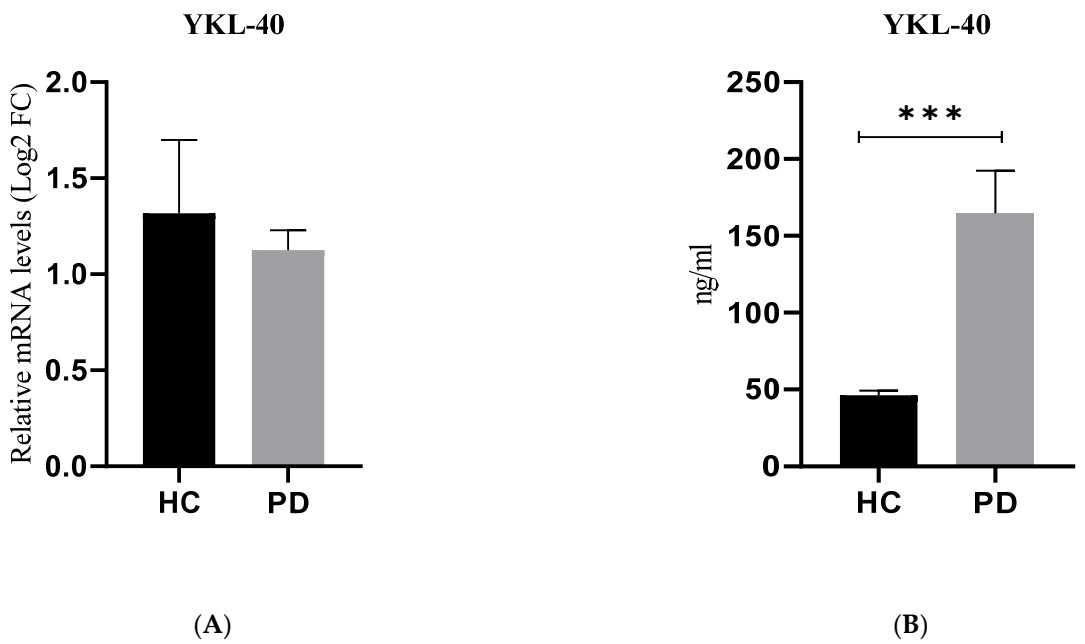

(A)                                                                                          (B)

**Figure 2.** Gene (**A**) and protein (**B**) expression of YKL-40 in Parkinson's disease (PD) and healthy controls (HC). Values are expressed as means ± S.E.M; *** $p < 0.001$.

### 2.4. Correlation Analysis

A possible relationship between YKL-40 protein levels and bioenergetic parameters was investigated. The glycoprotein was shown to be directly related to ATP production (R = 0.9, *p* = 0.012) and to basal respiration (R = 0.92, *p* = 0.008).

### 2.5. Receiver Operating Characteristic (ROC) Analysis

To evaluate the predictive value of the two statistically significant indicators—the spare respiratory capacity and the YKL-40 protein levels, ROC analysis was performed (Figure 3). Our results revealed Sensitivity%—75.08 and Specificity%—58.33 (Area—0.77, STDEV—0.10, *p* = 0.03) for the reserve respiratory capacity and Sensitivity%—80.0 and Specificity%—94.12 (Area-0.92, STDEV-0.04, *p* < 0.0001) for the plasma levels of YKL-40. The data show that YKL-40 plasma levels appear to be an excellent marker for the discrimination of PD from the HC. The spare respiratory capacity also differentiates the studied groups but is less important than the plasma levels of the glycoprotein.

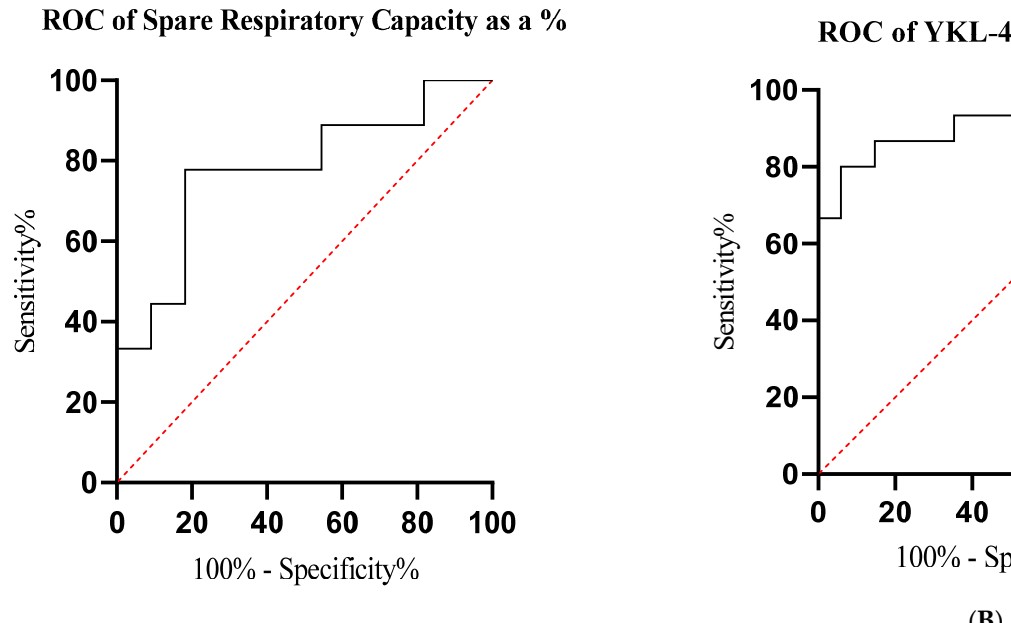

**Figure 3.** ROC analysis of Parkinson's disease patients (n = 18). (**A**) Spare respiratory capacity as %; (**B**) protein levels of YKL-40.

### 3. Discussion

In recent years, many reports have assumed mitochondrial dysfunction as a key factor in dopaminergic neuronal sensitivity in PD in both idiopathic and toxin-induced Parkinsonism [15–20]. This is explained by the fact that dopaminergic neurons have a high energy demand, and organelle dysfunction in patients promotes the loss of dopaminergic neurons in the substantia nigra pars compacta (SNpc). Gonzalez-Rodriguez et al. reported that the impairment of mitochondrial function simulates the pathogenesis of PD in animal models [15]. This evidence suggests that mitochondrial dysfunction occurs before the onset of PD. Therefore, one of the key challenges is the discovery of biomarkers that identify the early stages of the disease. The deeper understanding of the clinical–pathological mechanisms underlying the disease is fundamental for designing new strategies for therapeutic intervention in PD.

In the present study, we report high concentrations of plasma YKL-40 related to bioenergetic parameters. Data from previous investigations show a characteristic PD bioenergetic profile [21,22] with an increased reserve respiratory capacity compared to HC. The results obtained are consistent with other studies reporting changes in mitochondrial

activity in PD [21]. The difference between the examined groups is statistically significant and shows that the reserve capacity is sensitive to oxidative stress and is therefore a suitable marker for the detection of physiological abnormalities in neurodegenerative diseases [23] and in particular to dopaminergic disorders such as schizophrenia, restless legs syndrome (RLS), and PD [16].

Schirinzi et al. (2022) also reported increased reserve capacity and maximal respiration in PD that correlated with the severity of motor impairment and with CSF Aβ42 levels [21]. The team hypothesized that mitochondrial respiration capacity may increase with PD progression. It is interpreted as a compensatory bioenergetic activation as a result of abnormal mitochondrial function [21]. However, we detected that the other bioenergetic parameters, which were reduced in the patient group (ATP-coupled respiration, coupling efficiency, maximal respiration and proton leak), were not statistically significant. This suggests that mitochondrial function is suppressed, as observed by Walter J et al. as well [23]. The possible explanation is that the observed abnormalities in the investigated parameters are intrinsic to PD mitochondria or that their activity is related to other pathogenic processes leading to PD. It is known that mitochondrial dysfunction can provoke deposition of α-synuclein and formation of Lewy bodies, as well as a neuronal loss [24]. Altered astrocytic function also leads to α-synuclein accumulation, neuroinflammation, impaired mitochondrial metabolism, and oxidative stress [25]. Metabolic reprogramming results in less efficient production of ATP and ROS generation [26]. Neurons and differentiated neuronal cells are largely dependent upon oxidative phosphorylation and usually exhibit a substantial mitochondrial spare respiratory capacity, to rapidly meet the requirements of increased functional demands and stress [7]. Furthermore, respiratory capacity features the grade of toxicity, ischemia, and oxidative stress [7,27]. Schapira and Tolosa concluded that mitochondrial dysfunction in the brain of PD patients may exist before the point of diagnosis during the prodromal period [28]. As we expected, mitochondrial function was altered in our PD patients' group. These findings are in accordance with previous studies describing fluctuation in the OCR parameters in different neurological pathologies. It was found that the spare respiratory capacity in fibroblasts correlated with early neuropsychiatric symptoms in Alzheimer's disease [29]. This finding together with our data suggest that moderate changes in the activity of the electron transport chain in mitochondria can significantly affect the disease phenotype. Reduced mitochondrial ATP production was also found to correlate with the downregulation of the PD-related genes in human neurons [30]. Another factor involved in PD development, α-synuclein, was proven to modulate the activity of ATP synthase through interaction with its α-subunit [31].

Our data add new evidence to the existing literature and show the association of mitochondria with inflammation. Previous studies on skin fibroblasts and prodromal and early Parkinson's blood cells have reported the presence of compromised mitochondrial structure and function, oxidative stress, and increased levels of reactive oxygen species (ROS) coupled with decreased levels of the antioxidant superoxide dismutase [32].

In addition to mitochondrial changes, we report a correlation of bioenergetic indices and YKL-40 protein levels. This may be explained by the close relationship between inflammation and metabolism [33]. During the immune response, immune cells switch from a metabolic quiescence to an active phase. This transition is associated with metabolic reprograming from a catabolic to anabolic state and a concerted balance between ATP generation and its utilization [34–36]. We assume that YKL-40 might reflect different sides of the response to brain injury, such as neuroinflammation and brain damage. Many studies have reported that the levels of YKL-40 were associated with the number of cells involved in neurodegeneration and glial activation [37]. It was found that the concentration of YKL-40 was lower in patients with PD compared to those with atypical Parkinson's syndrome but still higher than the levels in the control group. No correlations with disease stages or severity were observed in this study [38].

Interestingly, the data regarding YKL-40 in PD remain controversial. Our results showed significantly increased plasma YKl-40 levels compared to HC without any impor-

tant change at the mRNA level. We could propose that YKL-40 expression might be regulated posttranscriptionally. In previous studies, a possible regulatory axis lncRNAs/miR-30e/YKL-40 was proven in systemic sclerosis [39]. It is quite possible that this glycoprotein is posttranscriptionally regulated in this pathology too.

High plasma levels of YKL-40 were detected in neurodegenerative dementias. Significantly elevated plasma YKL-40 in Creutzfeldt–Jakob disease (CJD) with a moderate potential to discriminate CJD cases from controls was determined. Additionally, YKL-40 concentrations appear significantly higher at late disease stages [40]. Furthermore, higher YKL-40 concentrations were related to the deterioration of cognitive abilities [37]. It was found that the high expression levels of YKL-40 were associated with an inflammatory brain profile in Alzheimer's disease [41]. Another research group determined that YKL-40 had the highest sensitivity and specificity in differentiating between healthy subjects and patients with mild dementia. It was concluded that only YKL-40 proved to be indicative for dementia progression [42]. On the contrary, other authors revealed lower levels of YKL-40 in patients with PD than in healthy controls or in other neurodegenerative conditions [43].

Along with the original results presented herein, however, there are some limitations in this study that could be addressed in future research. First, the number of patients and healthy individuals is insufficient; therefore, the data need further confirmation on a larger patient cohort. Second, a followup of the patients subtypes and the assessment of therapy is required. Although we used innovative and advanced technology to determine the mitochondrial status of living cells in real time, it is still an expensive approach, and it could be a limitation in routine clinical practice.

## 4. Materials and Methods

### 4.1. Patients and Controls

After signing a written informed consent in accordance with the instructions of the Ethics Committee at the Medical University of Plovdiv (Protocol № 6/2021) and following the principles of the Declaration of Helsinki, patients and healthy controls were recruited. It is a cross-sectional study that included 18 naïve PD patients and a control group (n = 7) of age-matched healthy individuals from a Bulgarian cohort. Venous blood was collected in an EDTA-Vacutainer monovette. The diagnosis was verified by a team of certified professionals from the Department of Neurology according to the internationally accepted criteria MDS-PD (Movement Disorder Society—Parkinson's Disease, 2015) (Postuma et al.; 2015). The severity of the symptoms was assessed by the UPDRS (Unified Parkinson's Disease Rating Scale) and the Hoen–Yahr scale. The full revised version of UPDRS contains 50 questions, divided into four parts: nonmotor aspects of experiences of daily living, motor experiences of daily living, motor examination, and motor complications. Each question is anchored with five responses. The total UPDRS score ranges between 0 (normal neurological examination) and 199 (the highest severity of the symptoms). The Hoen–Yahr scale is a short five-point scale that emphasizes the lateralization of symptoms and the patients' need for assistance in performing activities of daily living. The cases were recruited from the Department of Neurology, University Hospital "Kaspela", Medical University of Plovdiv, and amongst the patients referred to the Expert committee for diagnosis and treatment of Parkinson's disease at the University Hospital "Kaspela".

The control group included volunteers without a history or clinical data of neurological diseases. According to the exclusion criteria, subjects with acute/chronic infectious/inflammatory and autoimmune diseases were not involved in the study. Demographic and medical history information was also collected.

### 4.2. Isolation of Peripheral Blood Mononuclear Cells (PBMCs)

The peripheral blood was processed immediately, and PBMCs were isolated using Pancoll (Pan Biotech Cat # P04-60500) according to the protocol established by the manufacturer. Cells were cultured overnight in RPMI-1640 medium (Pan Biotech Cat # P04-22100) with 10% FBS, 1% penicillin/streptomycin in a cell culture incubator at 37 °C, 5% $CO_2$,

and high humidity. The number of cells and their viability were determined using an automatic counter "LUNA" (Logos Biosystems, Anyang, South Korea). Cells were diluted with RPMI-1640 to a final concentration of $2 \times 10^5$ cells per well and then plated in Seahorse microplates for analysis. Separated plasma was frozen and stored at $-80$ °C for subsequent analysis.

### 4.3. Metabolic Analysis in Real Time

Mitochondrial activity was measured using a Seahorse XFp analyzer (Agilent, Santa Clara, CA, USA). The advantage of our study is that the cells are processed immediately after their isolation, which limits the negative effect of freezing. PBMCs were cultured on poly-D-lysine coated wells at a density of $2 \times 10^5$ cells/well. To verify that the cells were evenly distributed in a monolayer, the wells were visualized using an inverted microscope (Nikon eclipse TS 100, NIKON, Amstelveen, The Netherlands).

### 4.4. Mito Stress Test

A mitochondrial stress test was performed according to recommendations of Agilent (Agilent, Santa Clara, CA, USA) and following the protocol established by the manufacturer. Before analysis, the growth medium was replaced with Seahorse XF medium (Agilent Seahorse, pH = 7.4), and PBMCs were incubated at 37 °C without $CO_2$. Each sample was tested in triplicate, and the results were averaged. The Seahorse analyzer measures the oxygen consumption rate (OCR) as an indicator of mitochondrial respiration. The first measurement determines the baseline OCR followed by OCR after injection of the following inhibitors: oligomycin (1.5 μM), carbonyl cyanide 4-(trifluoromethoxy) phenylhydrazone (1 μM) (FCCP), and rotenone (0.5 μM), according to the manufacturer's instructions.

The use of the indicated inhibitors allows measurement of ATP-coupled respiration (oligomycin inhibits ATP synthase), maximal respiration (FCCP, reveals maximal ETC capacity), reserve respiratory capacity (difference between maximal respiration and basal respiration), and non-mitochondrial respiration (rotenone inhibits complex I).

### 4.5. YKL-40 Gene Expression by qPCR

Total RNA from white blood cells was extracted using Trizol (Invitrogen, No. 15596026) according to the manufacturer's instructions. The concentration of RNA was measured on a NanoDrop Nucleic Acid Quantification (Thermo Fisher Scientific, Waltham, MA, USA). cDNA was transcribed from 2 μg RNA using Genaxon GreenMasterMix (2×) (Genaxxon bioscienceGmbH, Ulm, Germany, Lot. No. M3023.0500) according to the kit's instructions. The primer combinations used for the longest RNA transcripts of CHI3L1 (YKL-40) were synthesized by Integrated DNA Technologies, (Leuven, Belgium). The sequences were as follows: Fw 5′-CTGCTCCAGTGCTGCTCT-3′, Rev 5′-TACAGAGGAAGCGGTCAAGG-3′. RNA normalization was performed using several internal controls: GAPDH (Fw 5′-AGG TCCACCACTGACACGTTG-3′, Rw 5′-AGCTGAACGGGATGCTCACT-3′), ACTINβ (Fw 5′-AGTGTGACG TGGACATCCGGA-3′, Rev 5′-GCC AGGGCAGTGATCTCCTCCT-3′), and hUBC (Fw 5′TCCTCAGGCAGAG-GTT GATCTT-3′, Rev 5′-GGACCAAGTGCAGAGG TGGACTCTT-3′) (Integrated DNA Technologies, Leuven, Belgium). The qPCR reactions were run on Rotor-Gene Q 600 (Qiagen, Hilden, Germany), and quantitative evaluations were performed by the $2^{-\Delta\Delta Ct}$ method. Each sample was analyzed in duplicate.

### 4.6. Detection of YKL-40 in Plasma

Plasma was isolated from peripheral blood after centrifugation; then, the supernatant was frozen at $-80$ °C. It was later used to measure the YKL-40 levels with an ELISA kit, specific for the protein encoded by the longer isoform (383 amino acids) (MicroVue YKL-40, Quidel, 9975 Summers Ridge Road, San Diego, CA 92121, USA, Lot. No. 088337), following the manufacturer's instructions. The detection limit was 10 μg/L. The intra-assay coefficient of variation (CV) was 5%, and the inter-assay CV was <6%. YKL-40 is stable

for at least 15 years in plasma samples stored at −80 °C [44]. The assay employs the sandwich-based ELISA method with optical density measured at 405 nm.

*4.7. Statistical Analysis*

Preprocessing of OCR values was performed using Wave Software2.6.3. (Seahorse Bioscience, Agilent, (available at https://www.agilent.com/en/products/cell-analysis/software-download-for-wave-desktop, accessed on 4 October 2023). Statistical analyses of data were performed with Graphpad Prism. Differences between normally distributed variables were evaluated for significance using the Welch's *t*-test for independent samples or the paired *t*-test for dependent samples and the Wilcoxon–Mann–Whitney test. The significance threshold was set at *p*-value < 0.05.

## 5. Conclusions

In conclusion, our study reveals new aspects of the relationship between the glycoprotein YKL-40 and mitochondrial function in PD. The bioenergetic and inflammatory changes affecting immune cells can illustrate the development of neuroinflammation and neurodegeneration. We consider that YKL-40 protein levels in combination with the dynamics in mitochondrial metabolism might serve as an additional tool to evaluate inflammatory activity and the clinical course of PD.

**Author Contributions:** Conceptualization, V.S. and M.K.; methodology, M.G.; software, M.G.; formal analysis, M.G.; clinical investigation, A.T.; investigation, M.K. and M.G.; writing—original draft preparation, M.G. and M.K.; writing—review and editing, V.S.; supervision, V.S. and A.T. All authors have read and agreed to the published version of the manuscript.

**Funding:** This research was financed by European Union-NextGenerationEU, through the National Recovery and Resilience Plan of the Republic of Bulgaria, project № BG-RRP-2.004-0007-C01.

**Institutional Review Board Statement:** The study was conducted in accordance with the Declaration of Helsinki and approved by the of the Ethics Committee at the Medical University of Plovdiv (Protocol № 6/2021).

**Informed Consent Statement:** Written informed consent was obtained from all subjects involved in the study.

**Data Availability Statement:** Data are contained within the article.

**Acknowledgments:** The authors acknowledge the work of Danail Minchev and Valentin Dichev in sample collection and processing.

**Conflicts of Interest:** The authors declare no conflict of interest.

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
