# Peer review of "YKL-40 and the Cellular Metabolic Profile in Parkinson’s Disease"

_ijms, doi:10.3390/ijms242216297_

Round 1
Reviewer 1 Report
Comments and Suggestions for Authors
Comments to the Authors regarding the manuscript entitled “YKL-40 and cellular metabolic profile in Parkinson's disease (IJMS-2672705)”.
Dear authors,
Your study embarks on an exploration of the relationship between YKL-40 protein levels and cellular metabolic profiles in Parkinson's disease (PD). The endeavor to determine the correlation between cellular bioenergetics and YKL-40 levels in peripheral blood mononuclear cells (PBMCs) from PD patients is commendable. Your results indicate a potential alteration in mitochondrial function in PD patients, as showcased by an elevated spare respiratory capacity. Notably, YKL-40 protein levels appear to be significantly elevated in PD patients relative to the control group. The correlation established between YKL-40 levels and basal respiration and ATP production is intriguing. The insights you provide underscore the prospective utility of YKL-40 as both a biomarker and therapeutic target in PD. This highlights the paramount importance of grasping mitochondrial dysfunction and inflammation as potential pathogenic mechanisms. Your research bears significant relevance in clinical settings, offering a beacon of hope and understanding to PD patients and their families. Your diligent efforts and this subsequent scientific contribution are truly commendable. Well done.
Specific Comments:
Major Comments
1. The participant pool comprised 18 newly diagnosed Parkinson's disease (PD) patients juxtaposed with an age-matched control group (HC) consisting of only 7 individuals. The limited sample size may pose challenges to the broader applicability of your findings. Furthermore, the PD patients were in the nascent stages of their disease progression. Such a restricted disease progression scope might further constrain the universality of the results.
2. The manuscript lacks demographic details about the participants, such as gender, duration since diagnosis, medication regime, daily living abilities, or the disease stage. A potential homogeneity in the sample could curtail the findings' applicability to a diverse demographic.
3. Employing a cross-sectional study design, wherein data is gathered at a singular point in time, constrains the ability to infer causal relationships. It merely offers a snapshot of the nexus between YKL-40 protein levels and mitochondrial function in PD.
4. The study doesn't encompass longitudinal data that would elucidate the temporal stability or evolution of YKL-40 levels and mitochondrial functionality. Successive, time-sequenced studies would offer a more granular understanding of these interrelationships.
5. While your research propounds the potential of YKL-40 protein levels and mitochondrial function fluctuations as biomarkers for PD, corroborative research in expansive cohorts is essential. The study doesn't furnish details about the sensitivity, specificity, or predictive prowess of these putative biomarkers.
Minor Comments
6. Could you elucidate where the study participants were recruited from?
7. On lines 78-79, there is a marked difference between twice and 1.44 times. Please specify the exact discrepancy in the spare respiratory capacity between the two groups. Is there a particular rationale for emphasizing the proton leak and basal respiration bars for PD in Figure 1?
8. Given that results are visually represented, it would be beneficial to provide exact statistical measures like mean and standard deviation in the main text for precise interpretation.
9. In Figure 2, is there a specific reason for accentuating the YKL-40 bar for PD?
10. Kindly revisit the content related to sensitivity penned down on lines 108-113.
11. In Figure 3, could you elucidate the significance of the employed symbols?
12. It would be prudent to commence with a concise summary of your pivotal results without reiterating your objectives.
13. Engage in a comparative analysis of your findings concerning established literature. How does your research either complement or challenge extant paradigms? Eschew generic statements and dive into the nuances of concordances or discrepancies. Avoid mere repetition of synthesized findings; instead, muse upon their ramifications, constraints, and any methodological intricacies that might have influenced the outcomes.
14. Ponder upon the clinical, academic, policy-driven, or research implications of your results. Refrain from making sweeping recommendations.
15. Identify and discuss the potential limitations of your study. It is not merely an exercise in listing constraints but an introspection into the most pertinent ones and their influence on your findings.
16. Reflect upon the subsequent steps warranted in this research domain. Abstain from making generic proclamations about the necessity for more research.
17. Ensure your discussion weighs in on the magnitude of effects, their import, and their resonance or discordance with prevailing theories. Contemplate the patient's perspective.
18. Confront the limitations inherent in your study stemming from aspects like methodology, participant count, experimental design, etc. Muse upon subsequent research trajectories this study might inspire.
Warm regards,
Reviewer.
Author Response
Dear Мrs/Мr,
Thank you for your recommendations and for helping us to improve the paper.
Our corrections are marked (red) in the main text.

Reviewer 2 Report
Comments and Suggestions for Authors
The authors investigated the relationship between mitochondrial function associated with the pathophysiology of PD and YKL-40. As a result, changes in mitochondrial activity were observed in PD patients, and an association was seen between YKL-40 protein levels and mitochondrial basal respiration and ATP production. These observations suggest that the interaction between YKL-40 and mitochondrial function may play an important role in PD.
Therefore, I believe that this study provides valuable insights that may be extremely useful when considering additional tools for monitoring the clinical progression of PD.
However, I felt that just a slight modification is needed as follows
In the Abstract, it's also necessary to briefly indicate what YKL-40 is.
In the Introduction, it's necessary to explain that YKL-40 is a glycoprotein.
You mentioned that 'Our results show an almost two-fold increased spare respiratory capacity in PD patients compared to HC.'(p. 3 Line 78) However, isn't it closer to 1.5 times?
I hope these comments are helpful.
Author Response
Dear Мrs/Мr,
Thank you for your recommendations and for helping us to improve the paper.

Round 2
Reviewer 1 Report
Comments and Suggestions for Authors
Dear, Authors
I noticed from your response that you've made some revisions to the manuscript. However, it seems there are still areas that require further adjustments based on the review commoents. Would you kindly revise those sections and resend the updated manuscript? Thank you.
Author Response
Thanks to the constructive suggestions of the reviewer, we tried to improve the article by correcting the indicated sections.

Round 3
Reviewer 1 Report
Comments and Suggestions for Authors
Comments to the Authors regarding the manuscript entitled “YKL-40 and cellular metabolic profile in Parkinson's disease (IJMS-2672705)”.
Dear author(s),
I appreciate the diligence you have applied in revising your manuscript. The effort you have invested is evident in the quality of your work. It is essential to articulate these efforts clearly within your manuscript, as it is a significant step in scholarly communication.
I have observed that the formats of panels a and b in Figure 2 are inconsistent. Please address this discrepancy to maintain the uniformity of the presentation.
I extend my best wishes for your continued research endeavors. Please show your strong job ethics.
Warm regards,
Reviewer.

Author Response
We deeply appreciate your professional attitude and expertise in reviewing our paper. Thank you for your dedication and positivity!
